# Exploring the Phytobeneficial and Biocontrol Capacities of Endophytic Bacteria Isolated from Hybrid Vanilla Pods

**DOI:** 10.3390/microorganisms11071754

**Published:** 2023-07-05

**Authors:** Guillaume Lalanne-Tisné, Bastien Barral, Ahmed Taibi, Zana Kpatolo Coulibaly, Pierre Burguet, Felah Rasoarahona, Loic Quinton, Jean-Christophe Meile, Hasna Boubakri, Hippolyte Kodja

**Affiliations:** 1QualiSud, CIRAD, Université Montpellier, Montpellier SupAgro, Université d’Avignon, Université La Réunion, F-34000 Montpellier, France; guillaume.lalanne-tisne@univ-reunion.fr (G.L.-T.); bastien.barral@cirad.fr (B.B.); ahmed.taibi@univ-reunion.fr (A.T.); czana82@yahoo.fr (Z.K.C.); meile@cirad.fr (J.-C.M.); 2Université de La Réunion, 7 Chemin de l’Irat, F-97410 Saint Pierre, France; 3Mass Spectrometry Laboratory-MolSys Research Unit, ULiège, 4000 Liège, Belgium; pburguet@uliege.be (P.B.); loic.quinton@uliege.be (L.Q.); 4École Supérieure des Sciences Agronomiques, Département IAA, Université d’Antananarivo, Antananarivo 101, Madagascar; rasoafelah@yahoo.fr; 5CIRAD, 7 Chemin de l’Irat, F-97410 Saint Pierre, France; 6Ecologie Microbienne, Université Claude Bernard Lyon 1, CNRS, INRAE, 69622 Villeurbanne, France; hasna.boubakri@univ-lyon1.fr

**Keywords:** vanilla endophytic bacteria, plant microbiology, PGPB, biocontrol activities, spectral imaging

## Abstract

In this study, 58 endophytic bacterial strains were isolated from pods of two hybrid vanilla plants from Madagascar, *Manitra ampotony* and *Tsy taitra*. They were genetically characterized and divided into four distinct phylotypes. Three were associated to genus *Bacillus* species, and the fourth to the genus *Curtobacterium*. A selection of twelve strains corresponding to the identified genetic diversity were tested in vitro for four phytobeneficial capacities: phosphate solubilisation, free nitrogen fixation, and phytohormone and siderophore production. They were also evaluated in vitro for their ability to biocontrol the growth of the vanilla pathogenic fungi, *Fusarium oxysporum f.* sp. *radicis vanillae* and *Cholletotrichum orchidophilum*. Three bacteria of phylotype 4, m62a, m64 and m65, showed a high nitrogen fixation capacity in vitro, similar to the *Pseudomonas florescens* F113 bacterium used as a control (phospate solubilizing efficiency respectively 0.50 ± 0.07, 0.43 ± 0.07 and 0.40 ± 0.06 against 0.48 ± 0.03). Strain t2 related to *B. subtilis* showed a higher siderophore production than F113 (respectively 1.40 ± 0.1 AU and 1.2 ± 0.1 AU). The strain m72, associated with phylotype 2, showed the highest rate of production of Indole-3-acetic acid (IAA) in vitro. Bacteria belonging to the pylotype 4 showed the best capacity to inhibit fungal growth, especially the strains m62b m64 and t24, which also induced a significant zone of inhibition, suggesting that they may be good candidates for controlling fungal diseases of vanilla. This competence was highlighted with spectral imaging showing the production of lipopeptides (Iturin A2 and A3, C_16_ and C_15_-Fengycin A and C_14_ and C_15_-Surfactin) by the bacterial strains m65 confronted with the pathogenic fungi of vanilla.

## 1. Introduction

Vanilla is the only orchid of significant economic importance as an edible crop. It is a tropical hemi-epiphytic plant that grows in a humid environment [1]. About 100 species define the genus *Vanilla* in the family *Orchidaceaes* [2], but only three species are specifically cultivated for their pods: *Vanilla planifolia,* which represents the dominant cultivated species [3], *Vanilla pompona* [4] and *Vanilla tahitensis* [5]. In the 1990s, hybrids were created and disseminated in the cultivation areas of Madagascar: *Manitra ampotony* (*V. planifolia* × *V. tahitensis*) and *Tsy taitra* (*V. planifolia* × *V. pompona*) [6,7]. *Manitra ampotony* has a high vanillin content and, unlike *V. planifolia*, mature pods do not show a dehiscence slit. Thus, drying and ripening stages of the pod can take place on the plant and do not require the numerous processing steps necessary to obtain marketable black pods [6,7]. The hybrid *Tsy taitra* also has a remarkable aroma potential, a high productivity potential, long pods, and high resistance to various diseases [8].

Vanilla harbors various endophytic microorganisms which could be used as markers of the terroir [9]. They live asymptomatically in plant tissues for at least part of their life cycle. These endophytic microorganisms (bacteria and fungi) use different mechanisms to exert various beneficial effects from plant growth promotion or biotic and abiotic stresses [10,11,12,13]. Among endophytic bacteria, a particular class called PGPB (Plant Growth Promoting Bacteria) are found in most plants and have specialized and acquired the ability to invade a host [14]. Thus, they are able to fully colonize the rhizosphere and phyllosphere [10,15,16]. These endophytic bacteria are involved in physiological mechanisms, such as the production of phytohormones [11,17] or the stimulation of specific activities of the host plant, leading to an increase in enzyme catalysis and improving water uptake to facilitate mineral nutrition (ion uptake, siderophore production) and seed germination [18,19]. Furthermore, endophytic microorganisms can occupy an ecological niche that overlaps with that of many plant pathogens, thereby stimulating host plant defenses. Endophytic bacteria are known to inhibit the growth of bacterial and fungal phytopathogens [14] through the production of antibiotics, siderophores, and hydrolytic enzymes [20]. These direct benefits may be enhanced by the stimulation of a systemic response in the host plant, leading to increased resistance responses against pathogens [16].

Vanilla does not escape fungal attacks. *Colletotrichum orchidophilum*, the agent of black spot disease, is considered as one of the major pathogens of vanilla, attacking the aerial organs of the plant, including the pod, and can be responsible for a reduction in pod production of 10–30% [21]. *Fusarium oxysporum f.* sp. *radicis-vanillae* is the fungal pathogen responsible for rot and stem rot (RSR), the disease that has the greatest impact on vanilla production worldwide [22] and has already been isolated from vanilla pods [9]. Bacteria competing in the environment with other micro-organisms possess direct antagonistic activities against pathogens through hyperparasitism or antibiosis and others through competition for nutrients [23,24]. However, direct antagonists which actively produce a broad spectrum of antimicrobial metabolites are considered most effective against competitors, allowing advantages for antibiotic-producing microorganisms in resource-limited environments [25]. Several studies have been carried out on the phytobenefic properties of orchid endophytes [26]. Nevertheless, few studies have addressed the subject of vanilla endophytes [27,28,29], and none have focused on the bacteria hosted by the vanilla pod, whereas the bacteria isolated from this organ could have remarkable properties. The isolation of bacteria from vanilla pods that possess antagonistic activities against the above-mentioned vanilla pathogens is a crucial step for the development of adequate and effective biocontrol products against these diseases.

The present study aims (i) to isolate and purify endophytic bacteria from green pods of the hybrids *Manitra ampotony* and *Tsy taitra*, (ii) to study the diversity of its strains by 16S RNA gene sequencing and phylogenetic analysis, (iii) to evaluate their potential effectiveness in promoting plant growth, and (iiii) to evaluate their antagonistic activities against two phytopathogens of vanilla, *Colletotrichum orchidophylum* and *Fusarium oxysporum* f. sp. *radicis-vanillae*.

## 2. Materials and Methods

### 2.1. Isolation of Bacteria and Condition of Growth

Endophytic microorganisms were isolated from mature green pods of two accessions of vanilla hybrids from Madagascar: *Manytra ampotony* and *Tsy taitra*. Three pods for each accession were collected at the same time in June 2014 from three different plants grown close to each other at the vanilla plantation Ambohitsara in the Sambava region of north-east Madagascar (14°56′56.41″ S and 50°3′34.18″ E). The epidermis of the pods was sterilized using the protocol of Khoyratty et al. (2015) [9]. Briefly, collected organs were washed under running tap water for 15 min. The washed pods were then dipped for 10 s in 95% alcohol and thoroughly flamed on all sides for 3 s. The sterility of the pod surface was verified by placing the surface in contact with a sterile lysogeny broth (LB) medium agar plate and checking for the absence of microbial growth. Four 3–5 mm-thick slices of sterilized pod were placed in a lysogeny broth (LB) medium agar plate (1.5% agar). Colonies were subcultured several times on a fresh LB agar plate until single colonies were obtained. Bacterial strains were stored at −80 °C in cryotubes (Mast Cryobank, Bootle, UK).

### 2.2. DNA Extraction

For each strain, a bacterial pellet of 48 -h bacterial cultures (10 μL) was resuspended in 100 μL of RES buffer (Macherey Nagel^TM^, Düren, Germany) and supplemented with lysozyme (Euromedex, Souffelweyersheim, France, 20 mg·mL^−1^). After it was shaken and incubated for 10 min at room temperature, 200 μL of sodium dodecyl sulfate (0.1%) and 300 μL of phenol-chloroform-isoamyl alcohol (25:24:1) were added. The suspension was homogenized by successive pipetting and centrifuged at 18,000× *g* and 5 °C for 5 min. The aqueous phase was recovered and then supplemented with 300 μL of phenol-chloroform-isoamyl alcohol (25:24:1). After it was shaken and centrifuged at 18,000× *g* and 5 °C for 10 min, the aqueous phase was then supplemented with 1/10 of its volume of sodium acetate and 1 volume of isopropanol. The suspension was mixed by turning it several times and stored at −20 °C for 15 h. This suspension was centrifuged for 30 min at 18,000× *g* and 5 °C, and then the supernatant was removed. The DNA pellet was dried for a few minutes in the open air and resolubilized in 100 μL elution buffer (TRIS-HCl pH 8.5). The concentration and quality of the extracted DNA were estimated by a Nanodrop^®^ spectrophotometer (Thermo Scientific NanoDrop™ 1000 Spectrophotometer, Waltham, MA, USA). The bacterial DNA of the 58 strains was normalized to 20 ng·μL^−1^.

### 2.3. PCR Amplification and Sequencing of 16S rRNA Gene

PCR amplification of the 16S rRNA gene was performed using the bacterial universal primers: pA (5’-AGAGTTTGATCCTGGCTCAG-3’)/pH (5’-AAGGAGGTGATCCAGCCG-CA-3’) [30] and 27F (5’AGAGTTTGATCMTGGCTCAG-3’) [31] / 1378R (5’-CGGTGTGTACAAGGCCCGGGAACG-3’) [32]. Additional PCR with primers targeting the internal region of the 16S rRNA gene-COM1 (5’-CAGCAGCCGCGGTAATAC-3’)/COM2(5’-CCGTCAATTCCTTTGAGTTT-3’) [33,34] was performed in case we obtained poor quality or short sequences. PCR reactions were conducted in a total volume of 25 μL containing 2.5 µL of buffer (10×), 0.75 µL of MgCl_2_ (50 mM), 0.5 µL of dNTPs (10 mM), 2.5 µL forward primer (10 μM), 2.5 µL reverse primer (10 μM), 2.5 µL of Dimethylsulfoxide (DMSO), 0.625 U of Taq polymerase (Promega, Madison, WI, USA), 12.625 µL of H_2_O and 1 µL bacterial DNA (20 ng/µL). PCR reactions were incubated for 4 min at 94 °C, followed by 30 cycles of [50 s at 94 °C, 50 s at 55 °C and 90 s at 72 °C] with a final extension of 3 min at 72 °C. PCR products were electrophoresed on a 1.5% agarose gel (in Tris base, acetic acid and EDTA 1× buffer) at a voltage of 110 V for 30 min and visualized after ethidium bromide staining. The PCR fragment was then purified on a Nucleobond AX10000 column (Macherey Nagel TM) and the concentrations of DNA recovered were determined with a spectrophotometer (Thermo Scientific NanoDrop™ 1000 Spectrophotometer). The PCR products were sequenced by Microsynth France SAS using the same primers.

### 2.4. Sequence Analysis and Taxonomic Identification

The DNA chromatograms were checked utilizing SnapGene^®^ Viewer 6.0.2 software, and the assembly was performed with CAP3 software online (https://doua.prabi.fr/software/cap3, accessed on 28 June 2023) [35]. The sequences obtained were aligned with the EzBioCloud Database Update 7 July 2021 [36]. Alignments were performed using ClustalW, and phylogenetic trees were constructed using the “Maximum Likelihood” method associated each time with a bootstrap of 500 repetitions in molecular evolutionary Genetics Analysis (MEGAX) software [37]. All sequences were submitted to the GeneBank^®^ database.

### 2.5. Phosphate Solubilizing Efficiency

The phosphate solubilizing efficiency of the selected strains was determined by following the protocol of Mehta and Nautiyal [38,39]. In brief, bacterial strains were grown for 7 days at 28 °C with continuous agitation (140 rpm) in a NBRIP-broth medium containing bromophenol blue (BPB, Merck) as a pH indicator. *Pseudomonas fluorescens* (F113), a strain known as a plant growth-promoting agent [40,41], was used as a positive control, and a non-inoculated medium served as a negative control. All these experiments were reproduced at least three times with three replicates. At the end of the incubation period, the final OD_600_ values were subtracted from the negative control values with the following equation:(1)Decrease in color intensity=O.D. of control at 600 nm−O.D of the culture medium at 600 nm.

### 2.6. Siderophore Production

The selected bacterial strain’s ability for siderophore-producing was verified by universal Chromo Azurol S (CAS) (Merck) assay [42]. Prior to the experiment, the glassware was cleaned with 3 mol·L^−1^ hydrochloric acid (HCl) to remove iron and subsequently washed in deionized water [43]. The 24-h bacteria cultures were used as inoculum, and bacterial concentrations were adjusted so that the optical density at 600 nm (OD600) was 0.5 in NaCl solution (0.85%). Then, 10 µL of each suspension were spotted onto a modified double-layered chromo azurol S (CAS)-MM9 agar plate [44]. MM9 agar as the bottom layer served as a nonselective rich medium for bacterial growth. To measure the Fe-chelating function of siderophores, experiments were performed with CAS-blue agar (dye solution 10 mL) as the top layer. A CAS reagent was prepared according to Schwyn and Neilands [42]. Briefly, 121 mg CAS was dissolved in 200 mL distilled water and 10 mL of 1 mM ferric chloride (FeCl_3_,6 H2O) solution prepared in 10 mM HCl. Then, 145.8 mg of hexadecyltrimethylammonium bromide [HDTMA, Merck] was slowly added with constant stirring to the resulting dark-purple mixture. The pH of the resulting dark-blue solution was adjusted to 5, then 6.048 g of Piperazine-N,N′-bis ethanesulfonic acid (PIPES, Merck) was added slowly with constant stirring to the mixture. While the mixture was being stirred, the pH was adjusted to 6.8. The chrome azurol S (CAS) agar (0.9% *w*/*v*) solution was overlaid onto colonies of solid culture. All reagents in the indicator solution were freshly prepared for each batch of CAS-agar. The plates were incubated for 7 days at 28 °C. Sterile water was used as a negative control, and *Pseudomonas fluorescens* F11 was used as a positive control. Experiments were independently reproduced at least three times each with three replicates. The appearance of an orange halo in the CAS-agar was evaluated (Appendix A). The percentage of halo was determined according to Pinter et al., 2017 [45] by the equation:(2)The activity unit (AU) of siderophore production=((halo diameter−colony diameter))/colony diameter

### 2.7. Indole-3-Acetic Acid Production by Using Salkowski’s Reagent

To evaluate the capacity of bacterial strains to produce indole acetic acid (IAA), the production levels were determined on Yeast Extract Mannitol broth medium (YEM) amended with L-Tryptophan (2 mg·mL^−1^) and KNO_3_ (1%) according to the Vincent method [46] with modifications. The broth medium was inoculated in triplicate with 24-h bacteria cultures, and bacterial concentrations adjusted so that the optical density at 600 nm (OD600) was 0.1. The cultures were incubated in the dark for 7 days at 28 °C, and then 1 mL of each culture was centrifuged at 5000 rpm for 30 min. Supernatant (100 µL) of each bacterial culture was added in separate wells of a microplate, followed by the addition of 100 µL of Salkowski’s reagent [50 mL perchloric acid, 35% HClO_4_, and 1 mL of 0.5 M ferric chloride FeCl_3_], and incubated at 37 °C for 30 min. A pink coloration confirmed the presence of IAA in the supernatant, which was quantified using a spectrophotometer at 535 nm. To assess whether the auxin production of the strains depends on the presence of L-Tryptophan and KNO3, cultures in the YEM broth without L-Tryptophane and KNO_3_ were used as controls. Sterile water inoculation was used as a negative control.

### 2.8. Bacterial Anaerobic Growth in Nitrogen-Free Solid Medium

To determine the nitrogen (N)-fixing capacity of selected strains, the 24-h bacteria LB cultures were centrifuged and washed with NaCl 0.85% (3 times). Bacterial concentrations were adjusted to an optical density at 600 nm (OD600) of 0.1 in NaCl solution, and then incubated in vials with semi-solid NFb medium [47,48] (components per liter: 5 g malic acid, 0.5 g K_2_HPO_4_, 0.1 g NaCl, 0.2 g MgSO_4_·7H_2_O, 0.02 g CaCl_2_·2H_2_O, and 2 mL of bromothymol blue (0.5% in 0.2 N KOH solution; pH indicator), 4.5 g KOH (pH control)), supplemented with 2 mL micronutrient solution (components per 100 mL: 0.004 g CuSO_4_·5H_2_O, 0.12 g ZnSO_4_·7H_2_O, 0.14 g H_3_BO_3_, 0.1 g Na_2_MoO_4_·2H_2_O, 0.117 g MnSO_4_·H_2_O), 1 mL of vitamin solution (0.1 g.L^−1^ biotin, 0.2 g.L^−1^ pyridoxine) and 4 mL of Fe-EDTA solution (1.64% (*w*/*v*)) and solidified with 1.8 g Oxoid agar-agar, pH was adjusted to 6.5–6.8 with NaOH. Each plate was incubated for 5 to 7 days at 30 °C. The non-inoculated medium served as a control. Nitrogen-fixing capacity is visually evaluated by the presence and the thickness of the formation.

### 2.9. Antagonism Assay

The antifungal activity of the selected bacterial strains was evaluated in vitro against two pathogens of vanilla, *Colletotrichum orchidophilum* (BS11) [21] and *Fusarium oxysporum* sp. *Vanillae* (Fo72a) [22]. For the confrontation test, a LB liquid preculture of each bacterial strain was prepared for 24 h at 33 °C under mild shaking. The concentrations of 24- h LB bacterial culture were estimated by calculating the OD at 600 nm, and then each preculture was diluted to an OD_600_ of 0.1. The two fungi were grown in PDA medium petri dishes. A 0.5 cm-diameter chunk of agar containing a one-week culture of fungus was placed on the center of a PDA agar plate, and 5 µL of the diluted bacterial preculture was deposited at 2 cm from the fungus. A negative control with water and a positive control with 5 µL of cycloheximide (100 mg·mL^−1^) were realized. The plates were incubated at 27 °C for seven days. To estimate the inhibition rate of each bacterium against fungus, this formula was used:100 × [1 − (*GS*)/(*GS*) *negative control*](3)
where (GS) represents the growth surface area of the fungus on the half plate where the confrontation occurred, and the (GS) negative control refers to the average growth surface area of the fungus in the half plate of the negative control after seven days. All areas were outlined and calculated with imageJ software 1.52a [49]. The inhibition zone (IZ) was also measured (Figure 1). All experiments were independently reproduced three times each with three replicates.

### 2.10. Visualization and Identification of Antifungal Metabolites

#### 2.10.1. Mass Spectrometry Imaging and Data Processing

The bacterial strains used for these analyses were m65 and m72, which respectively belong to *Bacillus siamensis* and *Bacillus thuringiensis* species. The fungal strains were *Fusarium oxysporum* sp. *Vanillae* (Fo72a) and *Colletotrichum orchidophilum* (BS11). Three microliters of both the bacterial cell suspension (OD 600 nm = 0.1) and the fungal suspension (10^5^ conidia·mL^−1^) were spotted at 10 mm distance on a PDA medium diluted to ⅕ enriched with agar to reach a final agar concentration of 7 g·L^−1^. The plates were incubated at 30 °C for 48 h. The microbial samples grown on the agar medium were cut directly from the Petri dish and transferred to a microscope slide (VWR, Radnor, PA, USA) previously covered with a double-sided conductive copper tape (3M). This assembly was then dried at 39 °C for 2 h. For MSI, a HCCA matrix solution was prepared at 5 mg.mL^−1^ in 80/20 ACN/water doped with 0.1% TFA (Sigma-Aldrich, Overijse, Belgium). Then two times 10 layers of matrix were sprayed onto the slides with the SunCollect instrument (SunChrom, Friedrichsdorf, Germany). The first layer was sprayed at a flow rate of 10 μL·min^−1^. The flow rate was increased by 10 μL·min^−1^ after each layer until it reached 60 μL·min^−1^. MSI samples were acquired on Solarix XR 9.4T (Bruker Daltonics, Bremen, Germany) with a file size of 2M (FWMH ± 300,000 @ 400 *m*/*z*). The mass spectrometer was systematically mass calibrated from 200 *m*/*z* to 2500 *m*/*z* before each analysis with a red phosphorus solution in pure acetone spotted directly onto the slides covered with a double-sided conductive copper tape to reach a mass accuracy better than 0.3 ppm. FlexImaging v5 (Bruker Daltonics, Bremen, Germany) software was used for MALDI MS imaging acquisition, with a pixel step size for the surface raster set to 200 μm. For each mass spectrum, one scan of 20 Laser shots was performed at a repetition rate of 200 Hz. The LASER power was set to 14% and the beam focus was set to “small”. Finally, data processing was performed with SCiLS Lab 2016b (SCiLS, Bremen, Germany). Images shown were generated after total ion count normalization.

#### 2.10.2. Liquid Chromatography−Mass Spectrometry Analysis and Data Processing

The samples for TIMS-TOF analysis were prepared as follows: The agar plugs containing bacteria, confrontation zone, and fungus were immersed in 1 mL of acetonitrile/water/trifluoroacetic acid (70/30/0.1 *v*/*v*/*v*). The mix was stirred at 850 rpm and 20 °C for 5 h (Thermomix.). The supernatant was diluted 1:10 in water before analysis. The chromatographic separation was performed on a M-ClassACQUITY UPLC (Waters, Milford, MA, United States). A 3-min-long sample trapping step was first achieved on a reversed-phase (RP) ACQUITY UPLC M-Class Trap Column (Symmetry C18, 100 Å, 5 μm, 180 μm × 20 mm, Waters, Milford, USA) prior to releasing on a ACQUITY UPLC M-Class BEH C18 analytical column (100 Å, 1.8 μm spherical silica, 75 μm × 100 mm, Waters, United States). Water and acetonitrile both supplemented with 0.1% (v:v) of formic acid (FA) were used as eluents and mixed according to a 32 min-long gradient method. The flow rate was set at 0.6 μL·min^−1^. The mass detection was performed on a timsTOF spectrometer (Bruker, Bremen, Germany) with the following parameters: timsTOF, scan range *m*/*z*, 100–2200. The untargeted profiling data were acquired using Auto MS/MS. Experimental data were processed using Bruker Compass DataAnalysis 6.0.

## 3. Results

### 3.1. Isolation and Phylogenetic Identification of Bacterial Strains

The culture of bacterial phenotypes on the solid LB medium resulted in 58 different bacterial strains. These strains were associated by their 16S sequences with six species already described with more than 98.5% identity (Table 1). The six reference strains were the following species: *Bacillus subtilis*, *Bacillus siamensis*, *Bacillus inaquosorum*, *Bacillus albus*, *Bacillus thuringiensis gv. Thurigiensis* and *Curtobacterium oceanosedimentum*).

The 16S rRNA gene sequences were used to reconstruct the phylogeny of the 58 isolates. The resulting phylogenetic tree shows that the populations are divided into 4 phylotypes (Figure 2). Phylotype 1 is associated with the species *Curtobacterium oceanosedimentum*. Phylotype 2 is associated with the species *Bacillus thuringiensis gv. thuringiensis* and *Bacillus albus*. Phylotype 3 is associated with the species *Bacillus subtilis* and *Bacillus inaquosorum,* and finally, phylotype 4 is associated with the species *Bacillus siamensis*. The species composing the community of cultivable bacteria isolated from *Manitra ampotony* green vanilla pods belong to all four phylotypes, whereas those composing the community of bacteria isolated from *Tsy taitry* mature green pods belong only to phylotypes 3 and 4.

### 3.2. Screening of PGPB Capacities

A total of 12 strains were selected for further study corresponding to the diversity characterized above. All 12 bacterial isolates except m61 belong to the genus *Bacillus*. The M61 strain is identified as belonging to the genus *Curtobacterium* (Table 1) and is the only representative of phylotype 1 (Figure 2). The other 11 belong to genus *Bacillus* and were distributed in the three other phylotypes as follows: m72 represented phylotype 2; m62a, m62b, m64, m65, and t24 represented phylotype 4; and m67, t2, t5, t17a and t30 represented phylotype 3. For the capacity to solubilize phospate, based on Fisher’s least significant difference procedure for multiple comparisons, at *p* < 0.05, F113 had the highest capacity (OD600 nm = 0.48 ± 0.03), followed by m62a, m65, and m64, with an OD600nm, respectively, 0.50 ± 0.07, 0.43 ± 0.07and 0.40 ± 0.06 (Appendix A). The phylotype 1 isolate m61 had a negligible solubilisation index. M61 did not produce siderophore, but strain t2 in phylotype 3 showed the greatest capacity to produce siderophores (1.40 AU ± 0.1). Strains of phylotype 4 all showed a siderophore production capacity comparable to that of the reference strain *P. fluorescens* F113. All studied isolates produced IAA but m72, representing phylotype 2, had the highest production rate of IAA in the study with a OD535 of 1.30 ± 0.191.27 ± 0.25. F113 was not tested for IAA production. Finally, only the m62b isolate lacked the capacity to grow in a nitrogen-free medium, thus having the capacity to fix atmospheric nitrogen. In contrast, t17a and t5 isolates associated with phylotype 3 showed the highest capacity to grow in a nitrogen-free medium.

### 3.3. Antagonism Assay

The twelve strains were challenged in antagonistic cultures for seven days with two fungal pathogens of vanilla, *Colletotrichum orchidophilum* (BS11) and *Fusarium oxysporum* sp. *Vanillae* (Fo72a). Figure 3A shows the growth inhibition rates of *Colletotrichum orchidophilum* (BS11) induced by each bacterial strain co-cultured with the fungus after seven days. A Kruskal--Wallis test performed to compare the effect of the 12 strains on the growth of BS11 revealed a statistically significant difference in the inhibition rate of the different strains (Appendix A). Based on Fisher’s least significant difference for multiple comparisons at *p* < 0.05, there was a continuum of groups (de, ef, f) associating the strains m61, m72, m67, t17a, and t30 with the negative control, indicating that strain did not show any significant inhibition of the growth of strain BS11; average inhibition rates ranged from −5.2% to 6.8%. Strains m62b, m65, and t24 (group ab and abc) induced a clear inhibition of the growth of the fungus; the average inhibition rates in this group were, respectively, 30.1%, 30%, and 29.1%. Strains m64, m62a, t2, and t5 induced a slightly less significant inhibition with average inhibition rates ranging from 16.5% to 25%. All strains belonging to phylotype 4 showed the ability to inhibit the growth of BS11. The strains m61 and m72 belonging to phylotype 1 and 2, respectively, did not show any ability to inhibit the growth of BS11. However, the strains belonging to phylotype 3 are distributed in the two (negative and positive) groups; m67, t17a, and t30 did not show any capacity to inhibit the growth of BS11. On the contrary, t2 and t5 had average inhibition rates of 17.3% and 25%.

The confrontations of the 12 bacterial strains against *Fusarium oxysporum* sp. *Vanillae* (Fo72a) for seven days (Figure 3B) showed a similar pattern as that described above for *C. orchidolphilum* (BS11). A Kruskal--Wallis test performed to compare the effect of the 12 strains on the growth of BS11 revealed a statistically significant difference in inhibition rate of the different strains (Appendix A). Based on Fisher’s least significant difference for multiple comparisons at *p* < 0.05, strains m61, m72, t17a, and t30 are grouped in the negative control as no significant inhibition of the growth of strain Fo72a was observed. The positive control showed an extremely high inhibition rate (78.2%); F. oxysporum sp. Vanillae (Fo72a) was more sensitive to the antifungal action of cycloheximide than *C. orchidophilum* (BS11). There was a continuum of groups bcd, abc, ab, and c grouping strains with significant ability to inhibit the growth of Fo72a, namely m62a, m62b, m65, m64 and t24.The average inhibition rate for these strains ranged between 32.4% and 40%. Strains t2 and t5 (groups cde and de) had a slightly less significant inhibition with average inhibition rates of respectively 24.5% and 23.7%.

The inhibition zone between the bacterial and fungal colonies was measured for bacterial strains found to be competent to inhibit the growth of the fungus (Figure 4). Figure 4A shows the inhibition zones between the bacterial strains and *C. orchidophilum* (BS11) on PDA after seven days of co-culture. A Kruskal--Wallis test performed to compare the mean of the inhibition zones between the bacterial strains and BS11 revealed that there was a statistically significant difference in the inhibition rates of the different strains. Based on Fisher’s least significant difference for multiple comparisons at *p* < 0.05, the strains m62b, m64, and t24 are included in the same group, and the inhibition zone in this group is, apart from outliers, between 1.11 and 3.03 mm. The m65 isolate showed a versatile profile, and m62a, t2 and t5 were placed together in the same group which did not induce a zone of inhibition against BS11. The strains m62b, m64, t24, m62b, and m65 belonged to phylotype 4, but only m62b, m64, and t24 produced an inhibition zone in co-culture with BS11. Even if t2 and t5 strains (phylotype 3) could inhibit the growth of BS11, they did not produce any inhibition zone. The same trend is observed when bacterial strains are confronted with *F. oxysporum sp. Vanillae* (Fo72a) (Figure 4B).

### 3.4. Identification and Distribution of Bioactive Lipopeptides

The chromatographic analysis of m65 isolate revealed the presence of three families of lipopeptides (Figure 5A). Mass spectra of ions from each family displayed similar peptide sequences but with varying aliphatic chain lengths. The precursor ions with m/z masses of 994.6440, 1008.6596, 1022.6743, and 1036.6912 corresponded to C12-surfactin, C13-surfactin, C14-surfactin, and C15-surfactin, respectively. The fragmentation of the precursor ions allowed for the identification of the amino acid sequence of the surfactins, as confirmed by the peptide sequence Leu-Leu-Val-Asp-Leu-Leu (Figure 5D). Five compounds belonging to the iturin family were identified: iturin A-1 (*m*/*z* 1029.5372), iturin A-2 (*m*/*z* 1043.5532), iturin A-3 (*m*/*z* 1057.5691), iturin A-6 (*m*/*z* 1071.5841), and iturin A-8 (*m*/*z* 1085.6005) (Appendix A). The mass spectrum decomposition at the *m*/*z* 1043.5532 ion revealed the two parts of the iturin-specific peptide sequence: Glu-Asp-Tyr-Asn and Glu-Pro-Asn-Ser. The precursor ions with *m*/*z* masses of 1449.7905, 1463.8046, 1477.8203 corresponded to C15-fengycin A, C16-fengycin A, and C17-fengycin A, respectively. In contrast, no lipopeptides were detected in the chromatogram of m72 when grown in monoculture or in co-culture with a fungus.

Mass spectrometry imaging made it possible to visualize the spatial distribution of the lipopeptides produced by bacterial strains confronted with the pathogenic fungi of vanilla (Figure 6). The images clearly show a dividing line between m65 and the fungi unlike the images of m72 where the fungi cover the bacterial strain. The analysis of the mass spectrometry images reported similarities in the spatial distribution of the lipopeptides belonging to the same family which could be classified according to their relative intensity. Two lipopeptides of each family with the highest intensity have been represented (Figure 6A,B). The distribution of iturins was diffuse all around the bacterial colony and over the entire fungal growth inhibition zone. The distribution of the fengycin family produced by m65 was detected in a short halo around the colony of m65. The localization of surfactins varied depending on the fungus against which the bacterium grew. In the confrontation between m65 and *F. oxysporum f.* sp. *Vanillae* (Fo72a), surfactins are distributed around the colony with higher intensity towards the outer parts. Another pattern is observed in the confrontation between m65 and *C. orchidophilum* (BS11), where the surfactins are polarized on the confrontation zone. Mass spectrometry images confirmed the absence of lipopeptides for the strain m72 (Figure 6C,D).

## 4. Discussion

In this study, we identified a population of endophytic bacteria isolated from the mature pods of two vanilla hybrids and then measured their phytobeneficial potential. The two vanilla hybrids were *Manitra ampotony* (*V. planifolia × V. tahitensis*), which accumulated high levels of vanillin, and *Tsy taitra* (*V. planifolia × V. pompona*), which was resistant to fungal diseases. Although a greater number of bacterial strains were isolated from *Tsy taitra* pods (34 for a total of 58), they were distributed in only two of the four phylotypes identified, whereas the 24 strains isolated from *Manitra ampotony* pods were distributed in all four phylotypes (Table 1). Bacteria strains associated with the species of the genus *Curtobacterium* and *Bacillus* of phylotype 2 were only isolated from pods belonging to *Manitra ampotony*. However, the pods were collected at the same time from hybrid plants grown close to each other at the vanilla plantation Ambohitsara. This could suggest a relationship between the composition of the bacterial communities and the genotype of the host plant. It has been shown that there is a close relationship between the microbiome and the genotype of the host plant. This observation is in agreement with different studies [50,51,52,53] showing that the composition of the bacterial communities is more dependent on the genotype of the host plant than on environmental parameters such as terroir, indicating that the plant selects its own microbiome.

M61 was the only identified strain of phylotype 1 associated with the genus *Curtobacterium.* Several bacterial species belonging to the actinomycete class are recognized as endo-phytic bacteria exhibiting numerous PGP characteristics [54]. However, bacteria of the genus *Curobacterium* are also known to be plant pathogenic bacteria. The best known example is *Curtobacterium flaccumfaciens,* which causes disease in many species of *Fabaceae* [55,56,57]. So far, no studies have reported the *Curtobacterium* species as a pathogen of vanilla or other orchid species. In our study, m61 did not show any PGP trait in vitro or in the biocontrol capacity of pathogenic vanilla fungi. It is possible that m61 has a neutral interaction with the plant host. It is also possible that the LB culture medium used in the study did not favour any optimal development of m61. Indeed, many studies use more specific media for actinomycetes like the ‘Actinomycete Isolation Agar’ (AIA) [58].

In this study, Phylotype 2 was characterized for its phytobeneficial potential with m72. The closest taxonomic relative is *Bacillus thuringiensis gv. Thuringiensis. Bacillus thuringiensis,* abbreviated in its commercial form to *Bt*, is well known for its insecticidal properties and is still one of the most widely used biopesticides [59,60]. It appeared interesting to test a strain close to *B. thurigensis* for its antifungal capacities and PGP functions. Thus, the m72 isolate did not show any antifungal potential against the fungi responsible for the main vanilla diseases. These observations are consistent with the metabolic analyses where no lipopeptide could be identified in the conditions used in the present study. Regarding its PGP functions, the m72 strain did not show any particular competence in the assimilation of phosphate and nitrogen or in the production of siderophores but was the best producer of IAA in vitro from tryptophan. It has been shown that IAA-producing rhizobacteria participate favorably in the seed germination rate and plant growth [61,62]. Strain m72 appears to be an interesting candidate for further in vivo studies. 

Strains of phylotype 3 were associated with *B. inaquosorum* or *B. subtilis* based on 16S sequences similarity (Table 1). However, the phylogenetic tree in Figure 2 shows that while the 16S sequences of the reference strains of *B. inaquosorum* and *B. subtilis* were indeed very similar, the strains associated with these two bacteria (phylotype 3) appeared to be relatively distant. Until recently, *B. inaquosorum* was considered a subspecies of *B. subtilis*, but consistent genomic and biochemical data have ruled in favor of two different species [63]. This could indicate that the phylotype 3 bacteria could be closely related but distinct species of *B. inaquosorum* and *B. subtilis*. This hypothesis is reinforced by the variability of results observed within the phylotype 3 strains that were selected to test the PGPB traits (Table 2). Indeed, the t2 strain showed the highest level of siderophore production with the F113 control, while the t17a and t30 strains of the same phylotype exhibited reduced levels. Similarly, t5 and t17a were the best nitrogen-free fixing strains of the study, while the other strains of phylotype 3 showed a lower level of nitrogen fixation.

Strains t2 and t5 were the only bacterial strains of phylotype 3 that showed an ability to significantly decrease the growth of *F. oxysporum f.* sp. *Vanillae* (Fo72a) and *C. oxysporum* (BS11) (Figure 3). However, no inhibition zone between bacteria and fungi was observed (Figure 4). Although some strains of *B. subtilis* have been shown to be competent as biocontrol agents in other studies [64,65], strains of phylotype 3 close to *B. subtilis* did not show as much biocontrol capacity in our study.

Bacterial strains belonging to phylotype 4 appeared to form a homogeneous group based on both phylogenetic and functional characteristics. These strains produce weak indole acetic acid, fix moderate levels of free nitrogen (except for strain m62b), and produce siderophores. M62a, m64, and m65 strains exhibit the highest levels of phosphate solubilization efficiency with the control strain F113. Furthermore, they display a higher capacity to inhibit the growth of *Fusarium oxysporum f.* sp. *Vanillae* (Fo72a) and *C. orchidophilum* (BS11) as evidenced by the clear zones of inhibition observed in confrontation cultures with strains t24, m62b, m64 and m65. To further understand this inhibitory properties, metabolic studies were conducted. The results indicated that the lipopeptides (surfactin, fengycin, and iturin) produced by strain m65 played a crucial role in the biocontrol of pathogenic fungi [66]. These lipopeptides have been shown to possess broad-spectrum antifungal activity against various fungi, such as *Fusarium oxysporum, Phytophthora infestans,* and *Botrytis cinerea*. Surfactin is believed to exert its antifungal effects by disrupting the fungal cell membrane and causing cell lysis [67]. Fengycin and iturin are thought to disrupt the fungal cell membrane and inhibit the biosynthesis of ergosterol, a component of fungal cell membranes, thereby exerting their antifungal effects [68].

The bacterial strains that showed significant PGP capacities in this study should be further tested in vivo for their potential of phytostimulation. The selected candidates will be m62a for its ability to solubilise phosphate, t2 for its high siderophore production, m72 for its ability to produce IAA, and t5 for its ability to fix free nitrogen. These strains would be individually inoculated on model plants to characterize their effect on growth but also pooled together to test a formulation carrying all PGP capacities. The strains of phylotype 4 showed a strong ability to inhibit the growth of vanilla pathogens in vitro, as well as their ability to produce a wide range of lipoptides involved in inhibiting fungi and inducing systemic resistance in plants. Thus, the t24, m62b or m64 strains should be tested in vivo, not only on vanilla but also on other types of culture, as a natural phytoprotective agent.

## Figures and Tables

**Figure 1 microorganisms-11-01754-f001:**
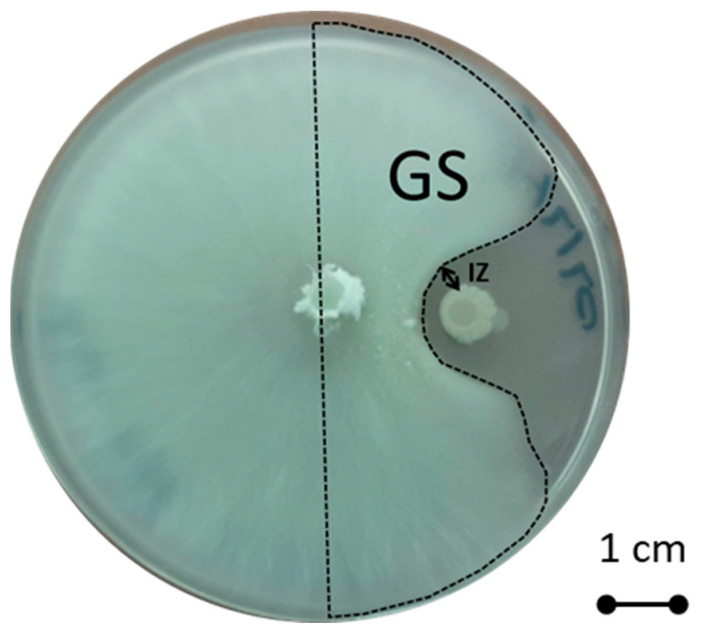
Antagonistic culture of *Colletotrichum orchidophilum* (BS11) with m64 bacterial strain. GS = Growth Surface area and IZ = inhibition zone.

**Figure 2 microorganisms-11-01754-f002:**
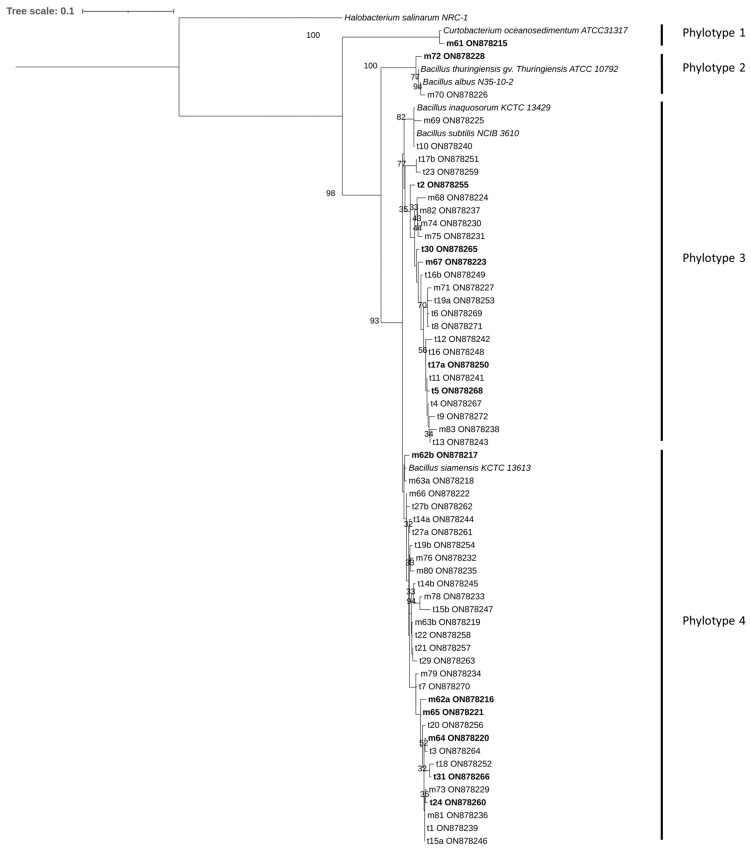
16S rDNA-based dendrogram showing the phylogenetic relationships between the different isolates and the most phylogenetically related species. The tree was rooted with the 16S rDNA sequence of *Halobacterium salinarum* as a reference outgroup. Only the ML bootstrap branches that support values greater than 30% are shown.

**Figure 3 microorganisms-11-01754-f003:**
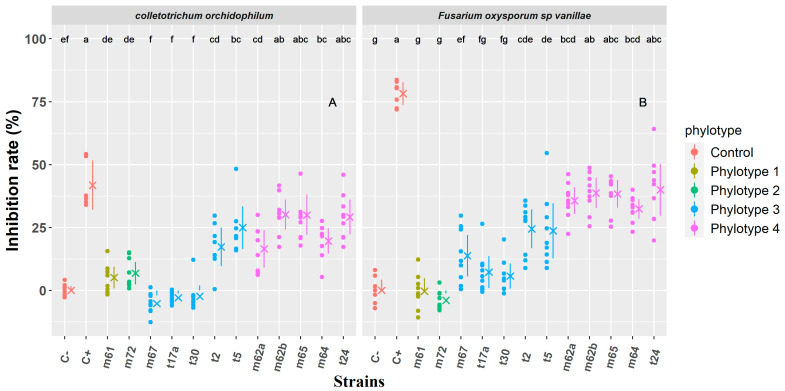
In vitro inhibition rate induced after 7 days by each bacterial strain on PDA medium on fungal growth of (**A**) *Colletotrichum orchidophilum* (BS11) and (**B**) *Fusarium oxysporum* sp. *Vanillae* (Fo72a). Each phylotype is defined by a color code. Positive (C−: growth of the fungus without confrontation) and negative (C+: cycloheximide) controls appear in gray. A dot is assigned to each measurement. Mean values are represented by crosses, and segments indicate standard errors. Different letters indicate that data are significantly different based on Fisher’s least significant difference for multiple comparisons at *p* < 0.05 with *p*_values adjusted with Bonferroni correction.

**Figure 4 microorganisms-11-01754-f004:**
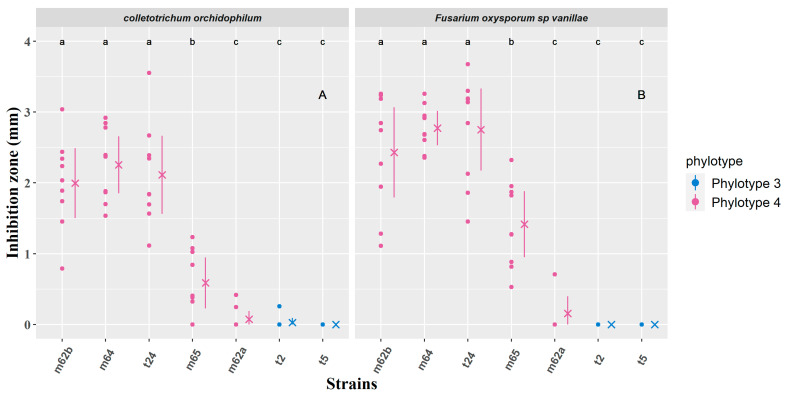
Mean inhibition zone size (mm) measured after seven days of co-cultures between the bacteria and: (**A**) *Colletotrichum orchidophilum* (BS11) and (**B**) *Fusarium oxysporum* sp. *Vanillae* (Fo72a). Each bacterial phylotype appears with a color code. In gray are the positive (C−: growth of the fungus without confrontation) and negative (C+: cycloheximide) controls. (**A**) A dot is assigned to each measurement. Mean values are represented by crosses, and segments indicate standard errors. Different letters indicate that data are significantly different based on Fisher’s least significant difference for multiple comparisons at *p* < 0.05 with *p*-values adjusted with Bonferroni correction (Appendix A).

**Figure 5 microorganisms-11-01754-f005:**
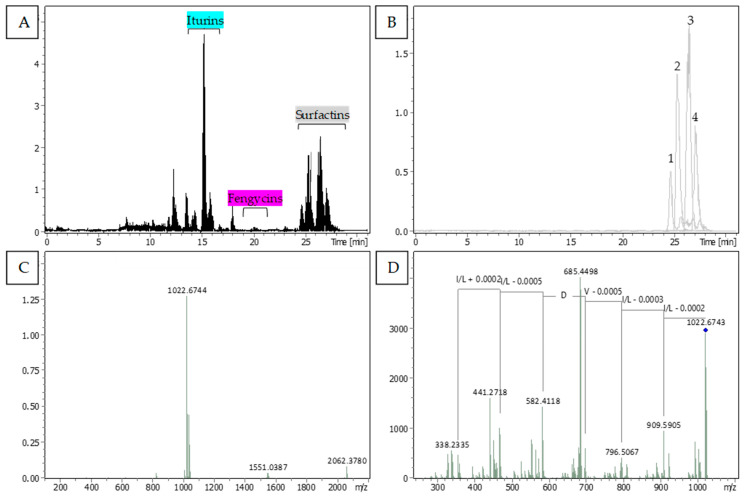
The TIMS-TOF analysis of compounds recorded in the inhibition area between m65 and *Fusarium oxysporum* sp. *Vanillae* (Fo72a). (**A**) Typical chromatogram with the distinct separation times of the three categories of identified lipopeptides: iturins (highlighted in blue), fengycins (highlighted in purple), and surfactins (highlighted in gray). (**B**) The extracted ion chromatogram of 994.6440 (1), 1008.6596 (2), 1022.6743 (3), and 1036.6912 (4) *m*/*z* representing the four present metabolites of surfactins. (**C**) MS spectrum represented the single charged M+H+ ion of surfactin 1022.67 *m*/*z*. (**D**) The MSMS spectrum ion at 1022.67 m/z and the fragmentation led to the peptide sequence analysis of surfactin. Similar processes of identification for iturin and fengycin families are presented in Appendix A.

**Figure 6 microorganisms-11-01754-f006:**
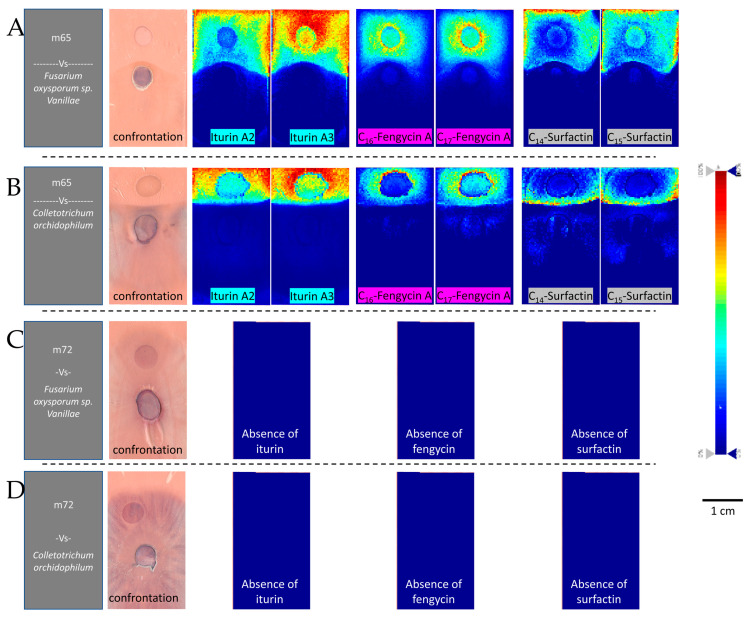
Mass spectrometry images showing the involvement of lipopeptides in *Bacillus spp.* antagonism against pathogenic fungi of vanilla. The images, from left to right, show the confrontation plate with the microorganisms, the distribution of various lipopeptides: iturins (highlighted in blue), fengycins (highlighted in purple), and surfactins (highlighted in gray), and intensity represented by a color gradient. (**A**,**B**) depict m65 against (**A**) *F. oxysporum sp. Vanillae* (Fo72a) and (**B**) *C. orchidophilum* (BS11). From left to right: picture of the confrontation plate, containing the microorganisms in culture transferred to the glass slide, previously covered with double-sided conductive copper tape; distribution of sodium adducts of Iturin A2 (*m/z* 1065.549, ± 25 mDa), Iturin A3 (*m*/*z* 1079.564, ± 25 mDa), C_16_-Fengycin A (*m*/*z* 1485.821, ± 25 mDa), C_17_-Fengycin A (*m*/*z* 1499.830, ± 25 mDa), C_14_-Surfactin (1044.676, ± 25 mDa) and C_15_-Surfactin (1058.689, ± 25 mDa). (**C**,**D**) panels show m72 against (**C**) *F. oxysporum sp. Vanillae* (Fo72a) and (**D**) *C. orchidophilum* (BS11). From left to right: picture of the confrontation plate, containing the microorganisms in the culture transferred to the glass slide, previously covered with double-sided conductive copper tape; no lipopeptides were detected in these images. The intensity of each compound is represented by a color gradient ranging from blue to red (0% to 100% intensity).

**Table 1 microorganisms-11-01754-t001:** Identification of the isolated bacterial strains. Strains in bold are those that were investigated in the following study for PGP and biocontrol functions. The prefix “t” in the strain name means that it was isolated from a mature green pod of *Tsy taitra,* and the prefix “m,” from a mature green pod of *Manytra ampotony*.

Strain	GeneBank Number	Closest Taxonomic Relative	% Identity
**m61**	ON878215	*Curtobacterium oceanosedimentum* ATCC 31317(T)	98.68% (1439 pb)
**m62a**	ON878216	*Bacillus siamensis* KCTC 13613(T)	99.31% (1455 pb)
**m62b**	ON878217	*Bacillus siamensis* KCTC 13613(T)	99.58% (1443 pb)
m63a	ON878218	*Bacillus siamensis* KCTC 13613(T)	99.52% (1453 pb)
m63b	ON878219	*Bacillus siamensis* KCTC 13613(T)	99..65% (1443 pb)
**m64**	ON878220	*Bacillus siamensis* KCTC 13613(T)	99.10% (1451 pb)
**m65**	ON878221	*Bacillus siamensis* KCTC 13613(T)	99.59% (1452 pb)
m66	ON878222	*Bacillus siamensis* KCTC 13613(T)	99.52% (1458 pb)
**m67**	ON878223	*Bacillus inaquosorum* KCTC 13429(T)	99.72% (1444 pb)
m68	ON878224	*Bacillus subtilis* NCIB 3610	99.25% (1459 pb)
m69	ON878225	*Bacillus subtilis* NCIB 3610	99.66% (1471 pb)
m70	ON878226	*Bacillus albus* N35-10-2(T)	99.39% (1474 pb)
m71	ON878227	*Bacillus subtilis* NCIB 3610	99.04% (1454 pb)
**m72**	ON878228	*Bacillus thuringiensis gv. Thuringiensis* ATCC 10792(T)	98.84% (1460 pb)
m73	ON878229	*Bacillus siamensis* KCTC 13613(T)	99.1% (1450 pb)
m74	ON878230	*Bacillus subtilis* NCIB 3610	99.18% (1458 pb)
m75	ON878231	*Bacillus subtilis* NCIB 3610	98.82% (1443 pb)
m76	ON878232	*Bacillus siamensis* KCTC 13613(T)	98.76% (1452 pb)
m78	ON878233	*Bacillus siamensis* KCTC 13613(T)	99.52% (1449 pb)
m79	ON878234	*Bacillus siamensis* KCTC 13613(T)	99.10% (1452 pb)
m80	ON878235	*Bacillus siamensis* KCTC 13613(T)	98.69% (1451 pb)
m81	ON878236	*Bacillus siamensis* KCTC 13613(T)	99.45% (1456 pb)
m82	ON878237	*Bacillus subtilis* NCIB 3610	99.38% (1459 pb)
m83	ON878238	*Bacillus subtilis* NCIB 3610	99.25% (1457 pb)
t1	ON878239	*Bacillus siamensis* KCTC 13613(T)	99.52% (1450 pb)
**t2**	ON878255	*Bacillus subtilis* NCIB 3610	99.31% (1458 pb)
t3	ON878264	*Bacillus siamensis* KCTC 13613(T)	99.31% (1453 pb)
t4	ON878267	*Bacillus subtilis* NCIB 3610	99.31% (1451 pb)
**t5**	ON878268	*Bacillus subtilis* NCIB 3610	99.31% (1451 pb)
t6	ON878269	*Bacillus subtilis* NCIB 3610	98.67% (1460 pb)
t7	ON878270	*Bacillus siamensis* KCTC 13613(T)	99.04% (1459 pb)
t8	ON878271	*Bacillus subtilis* NCIB 3610	99.25% (1457 pb)
t9	ON878272	*Bacillus subtilis* NCIB 3610	98.83% (1454 pb)
t10	ON878240	*Bacillus inaquosorum* KCTC 13429(T)	99.86% (14443pb)
t11	ON878241	*Bacillus inaquosorum* KCTC 13429(T)	99.31% (1448 pb)
t12	ON878242	*Bacillus subtilis* NCIB 3610	99.04% (1452 pb)
t13	ON878243	*Bacillus inaquosorum* KCTC 13429(T)	99.17% (1449 pb)
t14a	ON878244	*Bacillus siamensis* KCTC 13613(T)	99.25% (1459 pb)
t14b	ON878245	*Bacillus siamensis* KCTC 13613(T)	99.66% (1453 pb)
t15a	ON878246	*Bacillus siamensis* KCTC 13613(T)	99.38% (1459 pb)
t15b	ON878247	*Bacillus siamensis* KCTC 13613(T)	99.04% (1459 pb)
t16	ON878248	*Bacillus inaquosorum* KCTC 13429(T)	99.65% (1440 pb)
t16b	ON878249	*Bacillus inaquosorum* KCTC 13429(T)	99.17% (1452 pb)
**t17a**	ON878250	*Bacillus inaquosorum* KCTC 13429(T)	99.45% (1450 pb)
t17b	ON878251	*Bacillus subtilis* NCIB 3610	99.25% (1459 pb)
t18	ON878252	*Bacillus siamensis* KCTC 13613(T)	99.31% (1458 pb)
t19a	ON878253	*Bacillus subtilis* NCIB 3610	99.8.9% (1450 pb)
t19b	ON878254	*Bacillus siamensis* KCTC 13613(T)	98.84% (1460 pb)
t20	ON878256	*Bacillus siamensis* KCTC 13613(T)	99.52% (1460 pb)
t21	ON878257	*Bacillus siamensis* KCTC 13613(T)	99.38% (1459 pb)
t22	ON878258	*Bacillus siamensis* KCTC 13613(T)	99.38% (1457 pb)
t23	ON878259	*Bacillus subtilis* NCIB 3610	99.41% (1353 pb)
**t24**	ON878260	*Bacillus siamensis* KCTC 13613(T)	99.25% (1457 pb)
t27a	ON878261	*Bacillus siamensis* KCTC 13613(T)	99.52% (1460 pb)
t27b	ON878262	*Bacillus siamensis* KCTC 13613(T)	99.31% (1458 pb)
t29	ON878263	*Bacillus siamensis* KCTC 13613(T)	99.38% (1459 pb)
**t30**	ON878265	*Bacillus inaquosorum* KCTC 13429(T)	99.18% (1459 pb)
t31	ON878266	*Bacillus siamensis* KCTC 13613(T)	99.45% (1455 pb)

**Table 2 microorganisms-11-01754-t002:** Determination of phosphorus solubilization, IAA production, siderophore production and nitrogen fixation capacities of the different strains. The effect of the strains was evaluated by a non-parametric test (Kruskal--Wallis Rank Sum Test) and the highest values appear in bold based on Fisher’s least significant difference for multiple comparisons at *p* < 0.05 with p values adjusted with Bonferroni correction. The presented results are the means with standard error. The sign (−) indicates negative result, and (+) and (++) represent nitrogen fixation capacity demonstrated by the presence and thickness of the pellicle formed in nitrogen-free semi-solid media (NFb).

Code Name	Accession Number	Phylotype	Phosphate Solubilizing Efficiency (OD_600nm)_	The Activity Unit (AU) of Siderophore Production	IAA Production (OD_535nm_)	N2 Fixation
m61	ON878215	1	0.0 ± 0.0	0.0 ± 0.0	0.23 ± 0.06	+
m72	ON878228	2	0.05 ± 0.1	0.47 ± 0.08	**1.30 ± 0.19**	+
m67	ON878223	3	0.23 ± 0.04	0.93 ± 0.06	0.39± 0.13	+
t2	ON878255	3	0.32 ± 0.04	**1.40 ± 0.1**	0.16 ± 0.04	+
t5	ON878268	3	0.32 ± 0.02	0.87 ± 0.05	0.12 ± 0.01	++
t17a	ON878250	3	0.25 ± 0.03	0.73 ± 0.04	0.50 ± 0.11	++
t30	ON878265	3	0.22 ± 0.04	0.69 ± 0.05	0.42 ± 0.12	+
m62a	ON878216	4	0.50 ± 0.07	1.29 ± 0.1	0.15 ± 0.02	+
m62b	ON878217	4	0.25 ± 0.05	1.24 ± 0.1	0.09 ± 0.01	−
m64	ON878220	4	0.40 ± 0.06	1.01 ± 0.09	0.19 ± 0.5	+
m65	ON878221	4	0.43 ± 0.07	1.37 ± 0.14	0.13 ± 0.01	+
t24	ON878260	4	0.31 ± 0.05	1.17 ± 0.06	0.13 ± 0.02	+
F113	F113rif	(control)	**0.48 ± 0.03**	1.2 ± 0.1	−	+

## Data Availability

16SRNA gene sequences and related information are available on the GeneBank^®^ database. All the other relevant data are provided in the form of regular figures, tables, and Appendix A.

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
