# Peer review of "Exploring the Phytobeneficial and Biocontrol Capacities of Endophytic Bacteria Isolated from Hybrid Vanilla Pods"

_microorganisms, 2023, doi:10.3390/microorganisms11071754_

Round 1
Reviewer 1 Report
Reviewer comments:
Based on the 16S rRNA sequence alone, the analyzed bacteria should not be assigned to the species level. The sequence is too similar in some bacteria and it is impossible to distinguish between them. I recommend that the taxonomy of the analyzed strains be changed throughout the paper and remain at the genus level.
Have the analyzed data obtained from PGPR tests been analyzed for normality of distribution? Given the very large standard deviations, I conclude that the data did not have a normal distribution. In such a situation, other statistical tests should be sotos for comparative analysis and the data should be presented as median with confidence intervals.
The same situation occurs for antagonism test data.
The data obtained is very interesting. However, they require phylogenetic and statistical reanalysis.
Reviewer 2 Report
This is an interesting manuscript regarding the isolation identification and short characteristics of vanilla pods endophytic bacteria. The introduction is well written and the methods properly selected. The most interesting part concerns the use of MS imaging. There however some minor issues that should be corrected:
Over the methods, please provide the manufacturer data for the used reagents. It sometimes makes a huge difference which product is used for example for CAS media preparation and the size of the zones will be different depending on a lot of the components. Additionally, I would suggest using non inoculated media for IAA production assay as negative control since after longer incubation with the Salkowski’s reagent large concentration of L-Tryptophane can become slightly pinkish hue. Please chose a different method of antagonism activity comparison measurements when the measurement is strictly defined leaving no place for data manipulation. Please provide full L-Tryptophane statistics with test for data distribution and results for used test not only post-hoc test. Finally, the 16S sequencing is not enough to assign the isolates to the species level please limit it to genus level.
Concluding I find this manuscript suitable for the publication in Microorganisms after major revision regarding data presentation. I hope that my comment will help you to improve your manuscript, and I look forward to see it published.
In text comments:
Line 92: Please provide the exact location of material isolation.
Line 96: Please explain shortly the sterilization method. Solidified LB medium is called Lysogenic Agar you can use that name.
Line 102: Please provide manufacturer information for Lysozyme.
Line 105: Please explain how the sample was homogenized, please cite the used method and give producer information on the used reagents. Please give approximate times of incubation steps in time ranges.
Line 126: provide number of Units of polymerase.
Line 129 provide electrophoresis conditions gel percentage, voltage and time.
Line 188 Generally YEM medium contain traces amount of L-Tryptophane in yeast extract. Additionally, the ability of strains to produce IAA can differ from the L-Tryptophane and therefore strain A can produce more IAA than strain B on medium with less L-Tryptophane and the situation can reverse when more L-Tryptophane will be added. Therefore, I would suggest to use non inoculated medium as control for this experiments especially since L-Tryptophane can also give positive results after prolonged incubation.
Line 195 provide saline concentration.
Line 219 I believe it is to rough estimation the Growth area will be nowhere near the calculated Growth Surface, and there is no clear method of measuring R2. Please try to find other method of measuring inhibition maybe the minimal distance between bacterial colony and the fungus.
Line 271: 16S sequencing is currently considered as insufficient information for assigning strains to species level, based on this criterium strains can be assigned to genera.
Line 314: Please provide all statistics including the checking of criteria for the use of parametric test and the results of the test itself not only post-hoc test.
Line 335: Please use two-way test for this data, additionally please add the test result confirming that parametric test can be used.
Line 479 You are investigating the pod endophytic bacteria this may have a major influence on the diversity of microorganisms and why it is more species than environmental dependent. Moreover, the presented data do not allow for such comparison
Round 2
Reviewer 1 Report
Thank you for considering my previous review. The authors have followed all the instructions and now the work does not contain factual errors.
Reviewer 2 Report
Thank you for adequately addressing my concerns. I see that I misunderstood some parts of your manuscript I am glad you clarified that. My primary concern was the measurement of the antagonism and It has been corrected. I consider your publication to be acceptable for publication in its current form.